# COSINE SIMILARITY KNOWLEDGE DISTILLATION FOR INDIVIDUAL CLASS INFORMATION TRANSFER

## ABSTRACT

Previous logits-based Knowledge Distillation (KD) have utilized predictions about multiple categories within each sample (i.e., class predictions) and have employed Kullback-Leibler (KL) divergence to reduce the discrepancy between the student's and teacher's predictions. Despite the proliferation of KD techniques, the student model continues to fall short of achieving a similar level as teachers. In response, we introduce a novel and effective KD method capable of achieving results on par with or superior to the teacher model's performance. We utilize teacher and student predictions about multiple samples for each category (i.e., batch predictions) and apply cosine similarity, a commonly used technique in Natural Language Processing (NLP) for measuring the resemblance between text embeddings. This metric's inherent scale-invariance property, which relies solely on vector direction and not magnitude, allows the student to dynamically learn from the teacher's knowledge, rather than being bound by a fixed distribution of the teacher's knowledge. Furthermore, we propose a method called cosine similarity weighted temperature (CSWT) to improve the performance. CSWT reduces the temperature scaling in KD when the cosine similarity between the student and teacher models is high, and conversely, it increases the temperature scaling when the cosine similarity is low. This adjustment optimizes the transfer of information from the teacher to the student model. Extensive experimental results show that our proposed method serves as a viable alternative to existing methods. We anticipate that this approach will offer valuable insights for future research on model compression.

## 1 INTRODUCTION

In recent years, the advent of deep learning has led to remarkable progress in computer vision technologies, including image classification He et al. (2016a); Ma et al. (2018a), object detection Ren et al. (2015); He et al. (2017), and image segmentation Zhao et al. (2017); Long et al. (2015). However, these advanced models often require significant computational resources to achieve top performance, posing a challenge for real-world industrial applications Buciluǎ et al. (2006). To address these issues, various model compression techniques like model pruning Peste et al. (2022); Chen et al. (2023); Diao et al. (2023), quantization Qin et al. (2023); Koryakovskiy et al. (2023), and knowledge distillation (KD) have been introduced. Among these, KD stands out for its effectiveness and ease of use in a variety of computer vision tasks. This approach involves training a more compact student model using insights from a computationally-intensive teacher model, allowing the student to outperform its own self-training. This potent technique has firmly established itself as an invaluable asset for achieving efficient, yet high-performing solutions in tackling complex computer vision challenges Chen et al. (2017); Wang et al. (2020); Jiao et al. (2019); Peng et al. (2019).

Since its original introduction Hinton et al. (2015), KD has branched into two major methods: logits-based Zhao et al. (2022) and features-based Chen et al. (2021b). Logits-based methods train the student model using the teacher model's final output predictions, whereas features-based techniques use information from the teacher's intermediate layers. Because utilizing various features from teacher enables student to acquire a broader range of knowledge, features-based KD approaches often achieve higher accuracy than logits-based KD. However, they present practical challenges because, in some real-world situations, it may not be feasible to access the teacher model's intermediate layers due to safety and privacy issues. Therefore, our framework concentrates on the

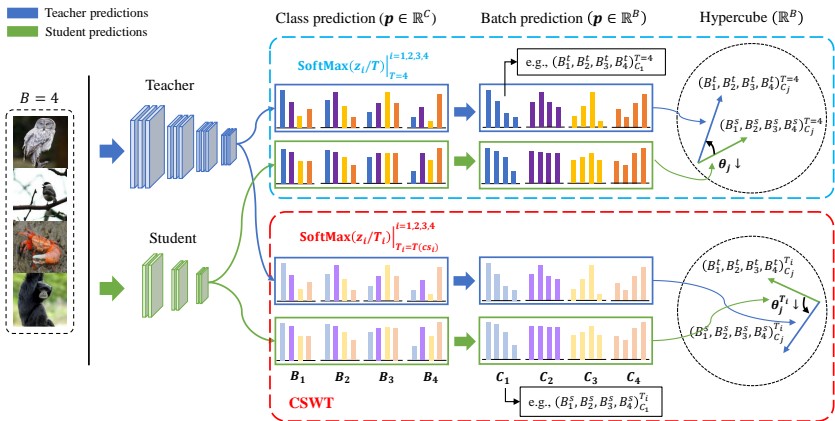

Figure 1: **Overview of the proposed method.** After the dataset passes through both the teacher and student models, two softmax processes are applied separately: a blue box using a fixed temperature ($T = 4$) and a red box employing a adaptive temperature ($T_i$, which we will explain as CSWT in the method section). During both processes, we rearrange the class predictions obtained after the softmax function into batch predictions. Subsequently, we consider these predictions as vectors within a hypercube and utilize cosine similarity between the teacher's and student's batch predictions to formulate the loss function.

logits-based approach, as it circumvents the need to access these intermediate features, making it more practicable for real-world deployment.

Even with the widespread adoption of KD, the student model still struggles to reach a comparable performance of the teachers. To bridge the performance gap between student and teacher models, we propose a novel logits-based distillation strategy that is both efficient and easy to deploy. Fig. 1 illustrates the entire procedure of our approach. After the training data passes through both the teacher and student models, our method utilizes two softmax processes to calculate the KD loss:(1) a constant temperature process, depicted by the blue dot box, and (2) an adaptive temperature process that varies for each sample, as indicated by the red dot boxes. In both processes, we reorganize the class predictions obtained after softmax into batch predictions. Then, we treat these batch predictions as a single vector and use cosine similarity to minimize the angle between the teacher and student vectors. This leverages the advantage of its scale-invariant properties for knowledge transfer, rather than employing methods that exhibit magnitude-dependent characteristics, such as Euclidean distance or Kullback-Leibler divergence. This manipulation decreases the biased prediction of student model and allows the student model to dynamically learn from the teacher's knowledge, rather than being restricted by the teacher's fixed distribution. As a result, our method can more effectively enhance the knowledge distillation process compared to existing approaches by fine-tuning predictions for non-target classes Li et al. (2022). This is supported by entropy analysis, which is detailed in the experiments section.

Furthermore, we suggest dynamic temperature scaling for individual samples, a method we refer to as "Cosine Similarity Weighted Temperature" (CSWT), which was mentioned as process (2). This approach adjusts temperatures based on the similarity between prediction of teacher and student model. The CSWT conveys more confident information by setting lower $T_i$ when the cosine similarity is high, and transfer richer information about the non-target class by setting higher $T_i$ when the cosine similarity is low. This effect has the advantage of providing more optimized knowledge for the student model than using a fixed temperature scaling factor.

Our contributions can be summarized as follows:

- We treat the predicted values from both the student and teacher models as vectors and employ cosine similarity to minimize the angle between these two vectors. Due to the scale-invariant property of cosine similarity, students learn more insights from teachers that encompass diverse possibilities.

- We suggest a Cosine Similarity Weighted Temperature (CSWT), which adjusts the temperature based on the cosine similarity value, enabling the student model to receive the most suitable information for each sample.

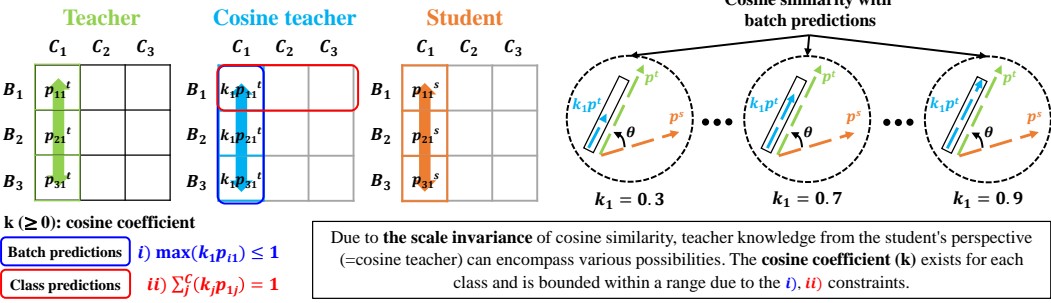

Figure 2: Illustrates the calculation of cosine similarity using our proposed batch predictions. The student model can learn the values of the cosine teacher during the learning process. This is because of the scale invariance of cosine similarity. The cosine coefficient (k) can take any value but is subject to two constraints. Since the cosine coefficient (k) can vary within a specific range, students can dynamically acquire the teacher's knowledge.

- Extensive experiments conducted on various datasets serve as evidence for the effectiveness of our proposed methods. Our approach achieves results comparable to that of the teacher model and, in some cases, even outperforms it.

## 2 RELATED WORK

### 2.1 KNOWLEDGE DISTILLATION

Knowledge distillation (KD) is an approach for compressing models, aiming to enhance both performance and efficiency. It involves transferring expertise from a high-performing yet inefficient teacher model to a less proficient but highly efficient student model. The various KDs available can be grouped into two categories: features- and logits-based distillation. Regarding features, Fit-Net Romero et al. (2014), RKD Park et al. (2019), and CRD Tian et al. (2019) have been proposed as methods that leverage intermediate features for more informative knowledge than logits. Recently, Review KD Chen et al. (2021a) has proposed a review mechanism that utilizes past features and has introduced ABF (Attention-Based Fusion) and HCL (Hierarchical Context Loss) to enhance performance. Previous research on handling logits include vanilla KD Hinton et al. (2015) and DKD Zhao et al. (2022), both of which introduced loss functions with the objective of diminishing the difference between the teacher's and student's final probability distributions. In particular, DKD proposes decoupling the loss function of vanilla KD into separate TCKD (Target Class KD) and NCKD (Non-target Class KD) components, enabling each part to affect performance independently. However, these investigations focus on transmitting knowledge among classes (i.e., class predictions), overlooking the importance of information exchange among batches (i.e., batch predictions). We emphasize the significance of batch predictions over class predictions. To achieve optimal alignment of batch predictions, we utilize the cosine similarity function, which is commonly used to measure similarity of vectors.

### 2.2 COSINE SIMILARITY

Cosine similarity is a fundamental metric that plays a crucial role in quantifying the similarity between vectors by measuring the cosine of the angle between them Nguyen & Bai (2010); Ye (2011). It is typically used in inner product spaces and is mathematically represented by dividing the dot product of vectors by the product of their Euclidean norms. Cosine similarity values range from 0 to 1, with 0 indicating orthogonality or a 90-degree angle between vectors. Conversely, a cosine similarity approaching 1 indicates a smaller angle between vectors, indicating increasing similarity. This versatile metric finds applications in various domains, including recommendation systems Melville & Sindhwani (2010), plagiarism detection El Mostafa & Benabbou (2020), and data mining Lahitani et al. (2016), to assess vector similarity in high-dimensional spaces. We conceptualize the batch predictions as vectors, representing positions within a hypercube with the batch size dimension (i.e., $p_{t,s} \in \mathbb{R}^B$ ). By designing the loss function to maximize cosine similarity between these vectors, we can develop a novel logits-based knowledge distillation (KD).

## 3 METHODOLOGY

This section covers the specifics of our knowledge distillation approach, which includes Cosine Similarity Knowledge Distillation (CSKD) and Cosine Similarity Weighted Temperature scaling (CSWT). These methods effectively transfer knowledge from the teacher to the student model.

### 3.1 BACKGROUND: COSINE SIMILARITY

The cosine similarity metric evaluates angle between two vectors in a multi-dimensional space and has found applications across diverse fields. One strength of this metric is that it is unaffected by the vectors' magnitude and relies solely on their direction. When considering two vectors, A and B, cosine similarity can be represented in terms of the inner product and the magnitude of each vector as follows:

$$\text{sim}(A, B) = \cos\theta = \frac{A \cdot B}{\|A\| \|B\|} \tag{1}$$

Although some previous KD research has utilized cosine similarity, its application has been restricted to measuring similarity between intermediate features vectors in features-based KD.

### 3.2 CSKD: COSINE SIMILARITY KNOWLEDGE DISTILLATION

After Hinton proposed vanilla KD, most logits-based KDs have utilized the Kullback-Leibler (KL) divergence, which measures the amount of information lost when approximating one probability distribution with another, to align the prediction score of students with those of teacher model. Numerous KL divergence-based KD methods have been proposed; however, a performance gap still exists when compared to the teacher model. To achieve comparable or even superior student model performance relative to the teacher model, we focus on the previously well-known fact that the teacher's information should be appropriately controlled and utilized by adjusting non-target class knowledge or softmax temperature scaling Li et al. (2022). Drawing inspiration from these, we propose a new loss function that leverages the scale-invariant property of cosine similarity as follows:

$$\mathcal{L}_{\text{CSKD}}\left(\boldsymbol{p}^s_{[:,j]}, \boldsymbol{p}^t_{[:,j]}, T\right) = 1 - \cos(\theta) = 1 - \frac{\boldsymbol{p}^s_{[:,j]} \cdot \boldsymbol{p}^t_{[:,j]}}{\left\|\boldsymbol{p}^s_{[:,j]}\right\| \left\|\boldsymbol{p}^t_{[:,j]}\right\|} \tag{2}$$

$$\boldsymbol{p}^{s,t}_{[i,:]} = \frac{e^{z_i^{s,t}/T}}{\sum_{k=1}^{C} e^{z_k^{s,t}/T}}, \tag{3}$$

where $z^{s,t}$ represent the logits of the student and teacher models, while $p^{s,t}$ denote the predicted probabilities of the student and teacher models, respectively (i.e., $p^s, p^t \in \mathbb{R}^{B \times C}$). Therefore, $p^s_{[:,j]}$ and $p^t_{[:,j]}$ mean the batch predictions of students and teacher model about $j$-th class, respectively.

Fig. 2 presents a conceptual representation of cosine similarity characteristics. We have chosen batch prediction over other existing KD methods that employ class prediction. This choice is driven by the ability to better exploit the scale-invariant attributes of cosine similarity. As depicted in Fig. 2, this scale-invariant property allows cosine teacher predictions to vary as long as they align with the direction of the teacher's predictions. Class prediction necessitates that the sum of predictions equals 1, making it less versatile, whereas batch predictions do not have this requirement. It demonstrates that students can dynamically acquire knowledge from teachers based on these conditions.

### 3.3 ANALYSIS OF COSINE TEACHER PREDICTIONS

We analyze variation in cosine teacher predictions $p^{cos}(k) = kp^t$ (Let, $k_{\min} \leq k \leq k_{\max}$), which is used to teach the student model, through entropy analysis. We defer the proofs of Lemma 4.1, 4.2 and Proposition 4.4 to Appendix.

**Lemma 3.1** (Maximum bound). *Given a cosine teacher prediction for particular class $j$ with $k$, $p_{[:,j]}^{cos}(k)$, $k_{max}$ can decreases as the batch size $B$ increases due to the constraint $\max p_{[:,j]}^{\cos}(k) \leq 1$.*

**Lemma 3.2** (Summation bound). *Given a cosine teacher prediction for particular sample $i$ with $k$, $p_{([i,:])}^{cos}(k)$, $k_{min}$ can increases and $k_{max}$ can decreases as the batch size $B$ increases due to the constraint $\sum_i^C p_i^{\cos}(k) = 1$.*

Under Lemmas 3.1 and 3.2, as $B$ increases, the $k$-range becomes narrower, leading to a decreased variation in cosine teacher predictions. Consequently, the cosine teacher prediction tends to remain relatively fixed when encountering the same samples during training (it is similar to KL-based KD).

**Assumption 3.3.** The range of $k$ is proportional to the entropy of the predictions generated by student model trained with $p^{\cos}(k)$, i.e., $\mathrm{range}(k) \propto \mathrm{entropy}\left(p^s\right)$

Assumption 3.3 is reasonable because, as the range of $k$ increases, the variation in cosine teacher predictions tends to increase. This suggests that the student model is exposed to diverse information even when encountering the same image during training, leading to an increase in entropy.

**Proposition 3.4.** *Under Lemmas 3.1, 3.2 and Assumption 3.3, the entropy of student prediction decreases as variation in cosine teacher predictions decreases,* i.e., $\mathrm{variation}(p^{cos}(k)) \propto \mathrm{entropy}\left(p^s\right)$

Proposition 3.4 implies that CSKD could result in higher entropy than KL-based KD (empirically Fig. 3) and increasing batch size could lead to lower entropy (empirically Fig. 4). Therefore, CSKD allows the student to acquire more information about non-target predictions, leading to higher performance.

### 3.4 CSWT: Cosine Similarity-Weighted Temperature Scaling

To provide students with a wide range of valuable information, we incorporate additional predictions by employing temperature scaling based on cosine similarity. The cosine similarity between the predictions of the student and teacher models for each sample can be calculated using Eq. 1 as follows:

$$cs_i = \frac{\boldsymbol{p}_{[i,:]}^s \cdot \boldsymbol{p}_{[i,:]}^t}{\left\|\boldsymbol{p}_{[i,:]}^s\right\| \left\|\boldsymbol{p}_{[i,:]}^t\right\|}. \tag{4}$$

When the cosine similarity for certain image $cs_i$ is high, we reduce the temperature scaling because it signifies a strong similarity between the student and teacher predictions for that sample. Conversely, when the cosine similarity is low, we increase the temperature scaling, interpreting it as indicating a significant dissimilarity between the student and teacher models. We achieve this by representing the temperature as a function of the cosine similarity, as follows:

$$T_i = (T_{\max} - T_{\min})\frac{cs_{\max} - cs_i}{cs_{\max} - cs_{\min}} + T_{\min} \tag{5}$$

$$\begin{cases} cs_{\max} = \max\left\{cs_1, cs_2, \ldots, cs_B\right\} \\ cs_{\min} = \min\left\{cs_1, cs_2, \ldots, cs_B\right\} \end{cases}, \tag{6}$$

where $i$ represents one sample of a batch size $B$, $T_{\min}$ and $T_{\max}$ are hyperparameters that define a temperature range. In our experiments, we set $T_{\min}$ to 2 and $T_{\max}$ to 6. As shown in Eq. 6, $cs_{\max}$ and $cs_{\min}$ represent the maximum and minimum values of the cosine similarity of the batch, respectively, which vary with each batch. Using this cosine similarity weighted temperature scaling, we can define an additional loss function as follows:

$$\mathcal{L}_{\mathrm{CSWT}}\left(\boldsymbol{p}_{[:,j]}^s, \boldsymbol{p}_{[:,j]}^t, T_i\right) = 1 - \cos(\theta) = 1 - \frac{\boldsymbol{p}_{[:,j]}^s \cdot \boldsymbol{p}_{[:,j]}^t}{\left\|\boldsymbol{p}_{[:,j]}^s\right\| \left\|\boldsymbol{p}_{[:,j]}^t\right\|} \tag{7}$$

$$\boldsymbol{p}_{[i,:]}^{s,t} = \frac{e^{z_i^{s,t}/T_i}}{\sum_{k=1}^{C} e^{z_k^{s,t}/T_i}}. \tag{8}$$

This loss function conveys ample dark knowledge concerning non-target predictions when dealing with images where there is a significant dissimilarity between the student and teacher models. Conversely, for images with a high degree of similarity, the loss function shifts its focus toward transmitting information specifically related to the target prediction. Consequently, this additional loss helps the student model acquire adaptive information from the teacher model, thereby enhancing the performance of the student model.

### 3.5 TOTAL LOSS FUNCTION

The total loss function of our framework including the cross-entropy loss and our loss with constant temperature and cosine similarity weighted temperature is formulated by

$$\mathcal{L}_{\text{Total}}\left(\boldsymbol{p}^s, \boldsymbol{p}^t, \ \Theta_s, \Theta_t, T, T_i\right) = \mathcal{L}_{CE}\left(\boldsymbol{p}^s; \Theta_s\right) + \alpha \left(\frac{1}{C} \sum_{j}^{C} \mathcal{L}_{\text{CSKD}}\left(\boldsymbol{p}_{[:,j]}^s, \boldsymbol{p}_{[:,j]}^t, T\right)\right)$$
$$+ \frac{1}{C} \sum_{j}^{C} \mathcal{L}_{\text{CSWT}}\left(\boldsymbol{p}_{[:,j]}^s, \boldsymbol{p}_{[:,j]}^t, T_i\right) \tag{9}$$

where $\Theta_s$ and $\Theta_t$ represent the parameters of the student and teacher models, respectively. The following experiments section validates that our suggested method is very simple and effective.

## 4 EXPERIMENTS

It is important to mention that **we repeated all experiments three times and presented the average results.** Implementation details are provided in the supplementary materials, which also include additional experiments covering time cost and hyper-parameter robustness.

We provide empirical evidence showcasing the effectiveness of our approach, which incorporates Cosine Similarity Knowledge Distillation (CSKD) and Cosine Similarity Weighted Temperature (CSWT), through experiments conducted on various datasets (***CIFAR-100*** Krizhevsky et al. (2009) and ***ImageNet*** Russakovsky et al. (2015)).

We performed experiments using well-known backbone networks, such as VGG Simonyan & Zisserman (2015), ResNet He et al. (2016b), WRN Zagoruyko & Komodakis (2016), MobileNet Sandler et al. (2018), and ShuffleNet Ma et al. (2018b), with various teacher-student model combinations.

The performance of the proposed method is evaluated in comparison to other knowledge distillation methods (KD, OFD, CRD, FitNet, DKD, SimKD, Multi KD, ReviewKD, and DPK) Heo et al. (2019); Hinton et al. (2015); Chen et al. (2021b); Tian et al. (2019); Romero et al. (2014); Zhao et al. (2022); Qiu et al. (2022); Jin et al. (2023); Chen et al. (2022).

### 4.1 CLASSIFICATION PERFORMANCE

Table 1 demonstrate the performance results of our method on the CIFAR-100 dataset. Regardless of whether the student and teacher model structures are the same or different, our method consistently delivers significant performance enhancements compared to other logits-based KD. Furthermore, it surpasses features-based KD, which leverages more abundant information from intermediate features. In certain cases, our models even outperform the teacher's performance.

We apply our method to the ImageNet dataset, utilizing both teacher and students models with identical and disparate architectures, allowing us to compare our method with previous logits-based KD and features-based KD. Table 2 presents the results, including both Top-1 and Top-5 accuracy, demonstrating that our method can achieve competitive performance over previous state-of-the-art KDs, even when dealing with noisy and large-scale datasets.

| | | WRN-40-2 | WRN-40-2 | ResNet56 | ResNet110 | ResNet32x4 | VGG13 |
|---|---|---|---|---|---|---|---|
| Types | Teacher | 75.61 | 75.61 | 72.34 | 74.31 | 79.42 | 74.64 |
| | Student | WRN-16-2 | WRN-40-1 | ResNet20 | ResNet32 | ResNet8x4 | VGG8 |
| | | 73.26 | 71.98 | 69.06 | 71.14 | 72.50 | 70.36 |
| Features | FitNet | 73.58 | 72.24 | 69.21 | 71.06 | 73.50 | 71.02 |
| | CRD | 75.48 | 74.14 | 71.16 | 73.48 | 75.51 | 73.94 |
| | OFD | 75.24 | 74.33 | 70.98 | 73.23 | 74.95 | 73.95 |
| | SimKD | 75.96* | 75.18* | 68.71* | 72.17* | 78.08 | 74.93 |
| | Review KD | 76.12 | 75.09 | 71.89 | 73.89 | 75.63 | 74.84 |
| Logits | KD | 74.92 | 73.54 | 70.66 | 73.08 | 73.33 | 72.98 |
| | DKD | 76.24 | 74.81 | 71.97 | *74.11* | 76.32 | 74.68 |
| | Multi KD | 76.63 | 75.35 | 72.19 | 74.11 | 77.08 | 75.18 |
| | Ours | **77.20** | **76.25** | **72.49** | **75.25** | **78.45** | **75.91** |

| | | WRN-40-2 | ResNet50 | ResNet32x4 | ResNet32x4 | VGG13 |
|---|---|---|---|---|---|---|
| Types | Teacher | 75.61 | 79.34 | 79.42 | 79.42 | 74.64 |
| | Student | ShuffleNet-V1 | MobileNet-V2 | ShuffleNet-V1 | ShuffleNet-V2 | MobileNet-V2 |
| | | 70.50 | 64.60 | 70.50 | 71.82 | 64.60 |
| Features | FitNet | 73.73 | 63.16 | 73.59 | 73.54 | 64.14 |
| | CRD | 76.05 | 69.11 | 75.11 | 75.65 | 69.73 |
| | OFD | 75.85 | 69.04 | 75.98 | 76.82 | 69.48 |
| | SimKD | 77.09* | 67.95* | 77.18 | 78.39 | 68.95 |
| | Review KD | 77.14 | 69.89 | 77.45 | 77.78 | 70.37 |
| Logits | KD | 74.83 | 67.35 | 74.07 | 74.45 | 67.37 |
| | DKD | 76.70 | 70.35 | 76.45 | 77.07 | 69.71 |
| | Multi KD | 77.44 | 71.04 | 77.18 | 78.44 | 70.57 |
| | Ours | **78.77** | **71.41** | **78.75** | **79.65** | **71.18** |

Table 1: Results on CIFAR-100 validation set using identical and disparate architectures. The best performance is marked in **bold**, and the second-best result is underlined. The results marked as * was not in their paper, so we conducted three new runs and then calculated the average.

| | Basic | | | Features | | | | Logits | | | |
|---|---|---|---|---|---|---|---|---|---|---|---|
| R50-MV2 | Teacher | Student | OFD | CRD | Review KD | DPK | KD | DKD | Multi-KD | Ours |
| Top-1 | 76.16 | 68.87 | 71.25 | 71.37 | 72.56 | 73.26 | 68.58 | 72.05 | 73.01 | **73.84** |
| Top-5 | 92.86 | 88.76 | 90.34 | 90.41 | 91.00 | 91.17 | 88.98 | 91.05 | 91.42 | **91.74** |
| R34-R18 | Teacher | Student | OFD | CRD | Review KD | DPK | KD | DKD | Multi-KD | Ours |
| Top-1 | 73.31 | 69.75 | 70.81 | 71.17 | 71.61 | 72.51 | 70.66 | 71.70 | 71.90 | **72.52** |
| Top-5 | 91.42 | 89.07 | 89.98 | 90.13 | 90.51 | 90.77 | 89.88 | 90.41 | 90.55 | **90.88** |

Table 2: Top-1 and Top-5 accuracy (%) on the ImageNet. In the row above, the teacher model is ResNet-50 and the student model is MobileNet-V2. In the next row, the teacher model is ResNet-34 and the student model is ResNet-18.

## 4.2 ENTROPY ANALYSIS

As explained in the Method section, our decision to incorporate cosine similarity into KD was motivated by the goal of leveraging the diverse range of cosine teacher predictions. In contrast to vanilla KD, which aims to mimic teacher predictions, cosine similarity operates within vector spaces, aligning teacher and student in the same direction, and has scale invariance. Consequently, this allows students to access a variety of predictions. To experimentally examine these diverse prediction possibilities, we used entropy as a tool.

In Fig. 3, we depict the entropy scores of our approach, employing cosine similarity, alongside vanilla KD, which relies on KL divergence, across multiple teacher-student pairs. The results clearly demonstrate that our method yields higher entropy values compared to vanilla KD, implying a broader range of potential outcomes than traditional KD techniques, enabling the student model to better learn non-target knowledge.

Fig. 4 visually illustrates the change in entropy as the batch size increases. As the batch size grows, the constraints imposed by maximum and summation bound become more stringent, resulting in a narrower range of $k$ values and a subsequent reduction in student prediction entropy. In contrast to vanilla KD, where the change in entropy is small across different batch sizes, our method exhibits a more pronounced shift in entropy.

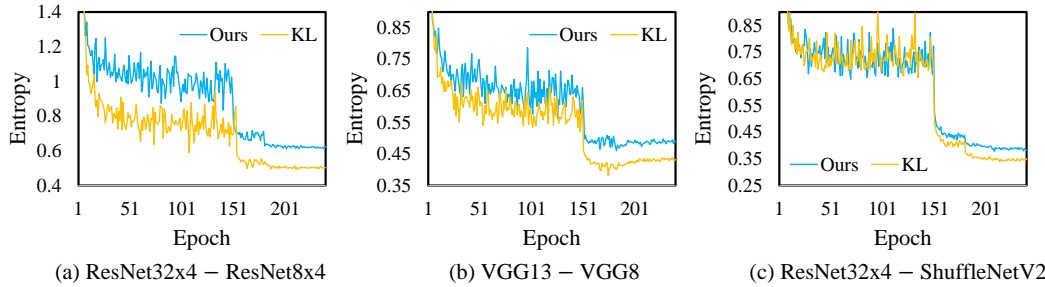

(a) ResNet32x4 − ResNet8x4      (b) VGG13 − VGG8      (c) ResNet32x4 − ShuffleNetV2

Figure 3: Compare the entropy of predictions extracted from the student model (for KL divergence and cosine distance loss, respectively) using the CIFAR100 test dataset. It can be seen that the average entropy is high when cosine distance loss is used. This shows that using cosine distance loss outputs slightly smoother predictions.

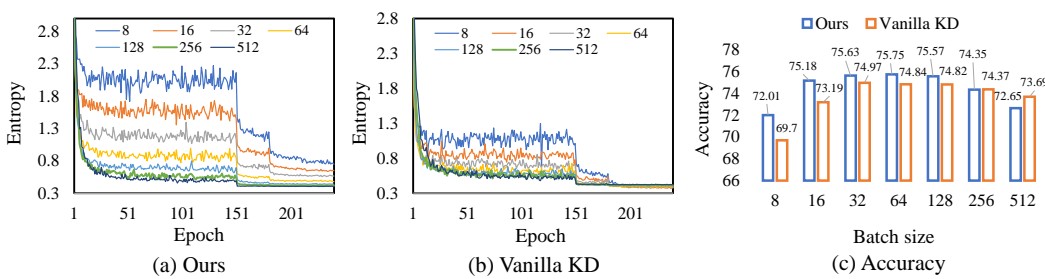

(a) Ours      (b) Vanilla KD      (c) Accuracy

Figure 4: Entropy change during training. We analyzed the model using the ResNet32x4-ResNet8x4 architecture. (a) is resulted by our method, while (b) is by vanilla KD. KD maintains consistent entropy post-convergence across different batch sizes, whereas our method exhibits increased entropy for smaller batch sizes even after reaching a lossy convergence.

Our study demonstrates that, in line with prior research Zhao et al. (2022); Jin et al. (2023), conveying the teacher model's knowledge is more effective when approached through appropriate moderation rather than a simple distillation of teacher information.

| CSKD | CSKD in multiKD | CSWT | Res32x4-Res8x4 | Res32x4-SV2. |
|------|-----------------|------|----------------|--------------|
|      |                 |      | 73.33          | 74.45        |
| ✓    |                 |      | 75.86          | 75.84        |
| ✓    | ✓               |      | 77.97          | 79.39        |
| ✓    | ✓               | ✓    | **78.45**      | **79.65**    |

Table 3: Ablation studies on Top-1 accuracy (%) on CIFAR-100 dataset for the proposed methods.

## 4.3 ABLATION STUDY

Table 3 shows the performance of the student model recorded while gradually applying each method. The teacher model is set as ResNet32x4, while the student model utilizes ResNet8x4 and ShuffleNetV2 to evaluate the effectiveness of our approach in both homogeneous and heterogeneous architectures. The performance in the first line represents the baseline with no application of our method, which corresponds to vanilla KD and exhibits the lowest performance among all scenarios. Replacing KL divergence with our CSKD results in performance enhancements of 2.53% and 1.39%, respectively. Furthermore, replacing MSE with our CSKD for batch-level and class-level alignment leads to additional improvements of 2.11% and 3.55%, respectively. Finally, implementing our CSWT boosts performance by 5.12% and 5.20% compared to vanilla KD. These findings highlight the significance of each component in our method for enhancing performance.

| Teacher | ResNet32x4 | VGG13 | VGG13 | ResNet32x4 | ResNet32x4 | ResNet50 |
|---|---|---|---|---|---|---|
| Student | ResNet8x4 | VGG8 | MobileNetV2 | ShuffleNetV1 | ShuffleNetV2 | MobileNetV2 |
| ReviewKD* | 75.63 | 74.84 | 70.37 | 77.45 | 77.78 | 69.89 |
| Ours** | 78.45 | 75.91 | 71.18 | 78.75 | 79.65 | 71.41 |
| **ReviewKD+Ours** | **78.99** | **76.31** | **72.17** | **79.10** | **80.33** | **71.91** |
| Δ* | **+3.36** | **+1.47** | **+1.80** | **+1.65** | **+2.55** | **+2.02** |
| Δ** | **+0.54** | **+0.40** | **+0.99** | **+0.35** | **+0.68** | **+0.50** |

Table 4: Orthogonality of ours methods with ReviewKD on CIFAR-100 datasets. $(\Delta*)$ signifies the difference between "ReviewKD" and "Review+Ours", while $(\Delta**)$ indicates the disparity between "Ours" and "Review+Ours".

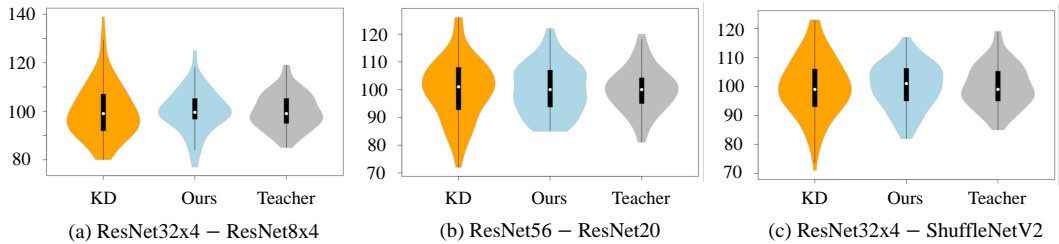

(a) ResNet32x4 − ResNet8x4    (b) ResNet56 − ResNet20    (c) ResNet32x4 − ShuffleNetV2

Figure 5: Compare the predicted classes extracted from the student models (for KL divergence and cosine distance loss respectively) using the CIFAR100 test dataset. Our method shows that it is more similar to the teacher's predicted class distribution than the method using KL divergence loss.

### 4.4 CLASS BIAS

Fig. 5 illustrates that our approach, as opposed to KL-based knowledge distillation, effectively mitigates class bias predictions. In the case of our method and the teacher's results, all class predictions tend to cluster around the value of 100. In contrast, with KL-based KD, these predictions can deviate significantly, reaching values of 140 or 70. This indicates that, during the distillation process, the KL-based approach tends to introduce biases of $+40$ or $-30$ for specific classes. These differing results between ours and KL-based method stem from the fact that our method employs cosine similarity in batch predictions, allowing the student to better capture non-target information while flexibly acquiring the teacher's knowledge.

### 4.5 ORTHOGONALITY TO FEATURES-BASED KD

Since our method doesn't necessitate external modules, it can be effortlessly assimilated into established features distillation methods. Table 4 demonstrates that our combined method notably enhances ReviewKD Chen et al. (2021b) (from 75.63 to 78.99) and consistently elevates our already powerful method to higher levels of performance.

## 5 CONCLUSION

In this paper, we introduce a novel logits-based knowledge distillation that utilizes the cosine similarity, a technique not employed in traditional logits-based knowledge distillation. By employing our CSKD (Cosine Similarity Knowledge Distillation), we effectively address the class bias problem, and its scale invariance allows the student model to dynamically learn from the teacher model. Furthermore, we integrate CSWT (Cosine Similarity Weighted Temperature) to enhance performance. Extensive experimental results demonstrate that our methods consistently outperform traditional logits-based and features-based methods on various datasets, even surpassing teacher models. Furthermore, our framework has demonstrated its ability to successfully integrate with existing feature-based KD method. We hope that our framework will find applications in a wide range of tasks in the future.

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

## A  APPENDIX

You may include other additional sections here.

