# OpenReview forum: "Cosine Similarity Knowledge Distillation for Individual Class Information Transfer"
_ICLR.cc/2024/Conference — Submitted to ICLR 2024_

### Official Review · Reviewer_j6ZX · 2023-10-31

**Soundness:** 3 good
**Presentation:** 3 good
**Contribution:** 3 good
**Rating:** 6
**Confidence:** 4

**Summary:**

This paper introduces a new Knowledge Distillation (KD) approach that relies on cosine similarity to effectively transfer knowledge about each specific class from the teacher model to the student model during the knowledge distillation process. By treating the predictions from student and teacher models as vectors, the method utilizes the scale-invariant property of cosine similarity to optimize student learning. The authors also introduced the "Cosine Similarity Weighted Temperature" (CSWT) technique to enhance the knowledge transfer efficiency.

**Strengths:**

1. This work employs cosine similarity for individual class information transfer, which is a departure from traditional KD techniques.
2. Efficient Learning from the student model. The authors of the paper leverage the scale-invariant property of cosine similarity to optimize the student's learning from the teacher.
3. With the introduction of the "Cosine Similarity Weighted Temperature" (CSWT) technique the student model refines the knowledge transfer process to obtain the most relevant information for every sample.

**Weaknesses:**

1. The proposed model depends on the batch size. This could be a limitation when the batch size that needs to be adjusted for reasons like memory constraints.

**Questions:**

1. Can this learning paradigm be used in an Out-of-distribution task or experiment?

---

> ### Author Response · Authors · 2023-11-15
> **Responses to the review of Reviewer  j6ZX**
>
> We sincerely appreciate the reviewer for their thorough and constructive comments. We are genuinely pleased that the novelty of the proposed method was recognized. In response to the reviewer's concerns, we are currently conducting additional experiments related to out of distribution. ***We would appreciate your review of these updated responses and kindly ask for your consideration in improving the score!***
>
> Thank you again for your time, feedbacks and patience.
>
> **Answer about W1)**
>
> As illustrated in Figure 4 of the main paper, our method demonstrates superior performance across various batch sizes, with the exception of very large batches (e.g., 512), when compared to vanilla KD. This observation implies that our approach, which achieves high performance even in small batch sizes, holds a distinct advantage over vanilla KD methods in real-world applications where memory constraints pose challenges.
>
>
> **Answer about Q1)**
>
> We are following your advice and conducting experiments on the out-of-distribution task. We are pleased to share our findings from the following experiment, in which the in-domain data is CIFAR-100, and the out-domain data is CIFAR-10.
>
> |Teacher-Student|Res34-Res18|WRN40_2-WRN16-1|WRN40_2-WRN40-1|WRN40_2-WRN16-2|
> |:---:|:---:|:---:|:---:|:---:|
> |KD|72.67|46.08|59.48|62.07|
> |Ours|**73.01**|**49.91**|**61.51**|**64.19**|
> |Difference|**+0.34**|**+3.83**|**+2.04**|**+2.13**|
>
> We conducted three runs for each architecture and reported the average. While this approach is tailored for standard KD tasks, we demonstrate its effectiveness even in OOD tasks, surpassing vanilla KDs. We plan to incorporate this into our future research. Your comments are greatly appreciated.

---

> > ### Comment · Reviewer_j6ZX · 2023-11-22
> > **Comments after rebuttal**
> >
> > Thank you for addressing my concerns and for conducting the experiment. I recommend this paper. Nice work.

---

### Official Review · Reviewer_RUAZ · 2023-11-01

**Soundness:** 2 fair
**Presentation:** 2 fair
**Contribution:** 3 good
**Rating:** 5
**Confidence:** 4

**Summary:**

The author sims to involve a cosine similarity KD loss based on a batch-level KD signal into student model learning. A combination of fix-temp softmax and a adaptive-temp softmax is also introduced in KD process. Comprehensive experiments including entropy analyses and ablation study are conducted.

**Strengths:**

S1. clearly introduce the research gap between existing KD techniques and the proposed

S2. Comprehensive experiments are conducted

**Weaknesses:**

W1. Cosine similarity KD loss is the key but the explanation is unclear

W2. presentation structure might be reorganized

W3. experimental results are not convincing due to unclear methodology

**Questions:**

C1. In Eq. (2) and (3), what does ":" mean? What's the difference between "[:,j]" in (2) and "[i,:]" in (3)? Does "[:,j]" mean by concatnating all vectors from 1 to j (if so, please refer to C3)? It said that p_s, p_t \in R^{B×C} which is a matrix. How to compute the consin similarity for two matrix based on Eq (1)?

C2. By the explanation in the below of Eq (3), it seems that the loss only focuses on one class j? I got the answer Yes from Eq (9). So it's suggested to make the definition clearer or reorganize the presentation order. It's better to explain all notations with a formal way without the assumption that audiences also understand what's presented.

C3. Though cosine similarity is scale-invariant, the significance of a cosine similarity value depends on vector length (i.e., the dimension of the vector). For a 200-d vector, 0.4 may be a relative large value. But for a 5-d vector, 0.7 does not mean two vectors are similar. For classification, the number of classes and  batch size are two factors to affect the vector length. That means, even for the same task, for difference batch size, the proposed would get different student models.

C4. The authors must take a long time to conduct extensive experiments in following sections to show the advantage of the proposed. However, concerns exist in the results due to unclear definitions and intuitions (please refer to C1 to C3)


==========

The authors response help my understanding on the methodology. I would like to raise my review score.

However, the concern is still on the batch-wise cosine similarity. Overall, I don't think it meet the acceptance bar.

---

> ### Author Response · Authors · 2023-11-15
> **Responses to the review of Reviewer RUAZ-Part-1**
>
> We appreciate the valuable feedback provided for our work. Over the past few days, we have diligently worked to address all the concerns raised. ***We kindly request the reviewer to reconsider and potentially raise the score, and we have outlined our responses to each point below.*** We will update the paper and appendices with the following your comments. Please feel free to let us know if you have any further concerns or questions to discuss.
>
> **Answer about Q1 and Q2**
>
> First of all, I apologize for any confusion regarding the notation. I explain this first and then share the amendments at the end. We adopted the meaning of '$:$' from Python array slicing. I am truly sorry for not being able to indicate this. The notation $i$ refers to the $i$-th element in the batch size, and the notation $j$ refers to the $j$-th class in the total number of classes.
>
> As a result, $p_{[i, :]}$ represents the distribution of prediction about single image sample, taking the form of vectors in $\mathbb{R}^{C}$. In contrast, $p_{[:, j]}$ denotes the distribution of prediction across samples for a single class, expressed as vectors in $\mathbb{R}^{B}$.
>
> Therefore, these expression allows us to easily compute the cosine similarity between the two vectors, not two matrix.
>
> Several concerns have been raised about the methodology, prompting the following correction. Your feedback on these adjustments would be greatly appreciated.
>
> The modification of Equations transforms it into the notation of ‘$tr$’ because the meaning of $[:, j]$ means a transpose function of $[i, :]$. (i.e., $p_{[i, :]}^{s, t}=p_{i}^{s,t} \in \mathbb{R}^{C}, p_{[:, j]}^{s, t}= p_{tr, j}^{s,t}\in \mathbb{R}^{B}$)
>
> The $p^s$ denotes the student's prediction, and the $p^t$ denotes the teacher’s prediction. $B$ refers to the size of the batch and $C$ refers to the number of classes.
>
> Previous formation eq (2):
>
> #### $L_{CSKD} ( p_{[:, j]}^s, p_{[:, j]}^t, T ) =$ $1-\cos (\theta) =  1- $$p_{[:, j]}^s\cdot p_{[:, j]}^t\over\vert p_{[:, j]}^s\vert\vert p_{[:, j]}^t \vert$
>
> After transformation eq (2):
>
> #### $L_{CSKD} (p_{tr, j}^{s}, p_{tr, j}^{t}, T)  =$ $1-\cos (\theta) = 1- $$p_{tr, j}^{s}\cdot p_{tr, j}^{t}\over\vert p_{tr, j}^s\vert\vert p_{tr, j}^t \vert$
>
>
> Previous formation eq (3):
>
> ### $p_{[i,:]}^{s,t}  =$ $e^{z_{i}^{s,t}/T}\over{\sum_{k=1}^{C}{e^{z_{k}^{s,t}/T}}}$
>
>
> After transformation eq (3):
>
> ### $p_{i}^{s,t}  =$ $e^{z_{i}^{s,t}/T}\over{\sum_{k=1}^{C}{e^{z_{k}^{s,t}/T}}}$
>
>
>
> Previous formation eq (4):
>
> ### $cs_{i}=$$p_{[i, :]}^s\cdot p_{[i, :]}^t\over\vert p_{[i, :]}^s\vert\vert p_{[i, :]}^t \vert$
>
> After transformation eq (4):
>
> ### $cs_{i}=$$p_{i}^s\cdot p_{i}^t\over\vert p_{i}^s\vert\vert p_{i}^t \vert$
>
> Previous formation eq (7) and eq (8):
>
> #### $L_{CSWT} ( p_{[:, j]}^s, p_{[:, j]}^t, T_{i} ) =$ $1-\cos (\theta) = 1- $$p_{[:, j]}^s\cdot p_{[:, j]}^t\over\vert p_{[:, j]}^s\vert\vert p_{[:, j]}^t \vert$
>
> ### $p_{[i,:]}^{s,t}  =$ $e^{z_{i}^{s,t}/T_{i}}\over{\sum_{k=1}^{C}{e^{z_{k}^{s,t}/T_{i}}}}$
>
> After transformation eq (7) and eq (8):
>
> #### $L_{CSWT} ( p_{tr, j}^s, p_{tr, j}^t, T_{i} ) =$ $1-\cos (\theta) = 1- $$p_{tr, j}^s\cdot p_{tr, j}^t\over\vert p_{tr, j}^s\vert\vert p_{tr, j}^t \vert$
>
> ### $p_{i}^{s,t}  =$ $e^{z_{i}^{s,t}/T_{i}}\over{\sum_{k=1}^{C}{e^{z_{k}^{s,t}/T_{i}}}}$
>
> Previous formation eq (9):
>
> #### $L_{Total} ( p^s, p^t, \theta_{s}, \theta_{t}, T, T_{i} ) =$ $L_{CE} ( p^s ; \theta_{s}) + \alpha($$1\over{C}$$\sum_{j}^{C}L_{CSKD}(p_{[:,j]}^s, p_{[:,j]}^t, T))+$$1\over{C}$$\sum_{j}^{C}L_{CSWT}(p_{[:,j]}^s, p_{[:,j]}^t, T_i)$
>
> After transformation eq (9):
>
> #### $L_{Total} ( p^s, p^t, \theta_{s}, \theta_{t}, T, T_{i} ) =$ $L_{CE} ( p^s ; \theta_{s}) + \alpha($$1\over{C}$$\sum_{j}^{C}L_{CSKD}(p_{tr, j}^s, p_{tr, j}^t, T))+$$1\over{C}$$\sum_{j}^{C}L_{CSWT}(p_{tr, j}^s, p_{tr, j}^t, T_i)$
>
> We provide an explanation for the notation $p_i^{s,t}$ because, when computing cosine similarity between $p^{s,t}_{tr, j}$ , the initial step involves obtaining $ p_i^{s,t} $ and subsequently transposing to derive
>
> $p^{s,t}_{tr, j}$
>
> For instance, for $p_i^{s,t}$, we pass the logits of the $i$-th image through the softmax function, and then we obtain the probability value for that image. Extending this to a batch size(let's say, with 64 images), we acquire the probability distribution for each of the 64 images ($p^{s,t}$), presented in the form of a $B \times C$ matrix. By transposing this matrix $(C \times B)$ and isolating the distribution for $j$-th class, we obtain
>
> $p^{s,t}_{tr, j}$
>
> —our desired output. This $p^{s,t}_{tr, j}$ can then be utilized for calculating cosine similarity.
>
>
> I hope the above modifications aid your understanding.

---

> > ### Author Response · Authors · 2023-11-15
> > **Responses to the review of Reviewer RUAZ-Part-2**
> >
> > **About Q3) with W1~W3**
> > Thank you for your thoughtful comments regarding the potential impact of vector length on cosine similarity values.
> >
> > Also, thank you very much for mentioning that the number of classes and batch size, which affect the vector length, are important factors in classification tasks.
> >
> > As a result, the proposed method does not perform concatenating all vectors, and we agree that student models may exhibit variations in performance for the same task due to differences in batch sizes. This is expressed experimentally in Figure 4, (C) in our paper.
> >
> >
> > We argue that the advantage of the proposed methodology is that the student can learn various predictions.
> >
> > This is because the existing KL-divergence aims to accurately match the student's prediction distribution and the teacher's prediction distribution. This requires ‘exact equality’ of logit values between students and teachers for a zero loss value. However, these strict matching criteria place overly strict constraints on students, limiting their ability to acquire flexible dark knowledge from teacher distributions, ultimately leading to poor student performance.
> >
> > To overcome these constraints, we introduce a loss function based on cosine similarity, which boasts scale-invariant properties. This distinguishes the approach by providing flexibility in the student logit without the need to exactly replicate the teacher distribution. As shown in Figure 2 of this paper, using cosine similarity increases the flexibility of dark knowledge acquisition by student distribution. Verification through entropy analysis experiments (see Figure 3 of this paper) highlights that our method exhibits higher entropy compared to existing KD methods. (It is worth noting that higher entropy corresponds to a greater amount of dark knowledge.)
> >
> >
> > **About Q4)**
> >
> > We appreciate the reviewer's consideration and feedback. The authors acknowledge that conducting extensive experiments in the upcoming section to demonstrate the strengths of the proposed approach may take some time. We are currently working on acquiring further comparative analyses, including visualizations such as "tSNE" and "logits correlation", which we will incorporate.
> >
> > We recognize the concern regarding unclear definitions and reliance on intuition, and we are committed to addressing these issues. The forthcoming revisions will aim to provide a more precise and well-defined analysis to alleviate any concerns about the results. Thank you for highlighting these points, and we will work diligently to enhance the clarity and rigor of our work.

---

> > ### Comment · Reviewer_RUAZ · 2023-11-22
> > **authors response addressed some of my concerns**
> >
> > Thank you very much for the response. The clarification help me understand the methodology. I would like to raise my review score.
> >
> > However, the concern is still on the batch-wise cosine similarity. Overall, I don't think it meet the acceptance bar.

---

### Official Review · Reviewer_9Nip · 2023-11-02

**Soundness:** 2 fair
**Presentation:** 2 fair
**Contribution:** 1 poor
**Rating:** 3
**Confidence:** 4

**Summary:**

The author proposes a method for knowledge distillation (KD) using cosine similarity, which yields favorable results on commonly used datasets such as CIFAR-100 and ImageNet. Despite its significant effectiveness, the paper still faces several issues that currently place it significantly below the acceptance threshold for ICLR.

Issues:
1. The author's concept of cosine distance is commendable but lacks a comparison and discussion with existing KD methods based on KL divergence, such as SHAKE [1] and DKD [2]. Incorporating relevant comparative analysis in the next version would be beneficial.
2. The paper does not adequately explore the relationship between the proposed method and existing KD techniques; it merely provides result comparisons. This leaves readers struggling to understand the unique significance of the proposed approach. A deeper discussion of the differences between these two types of KD methods is needed in the related work section.
3. The motivation behind the entire loss function is unclear. While the author introduces a temperature parameter (T) in Equation 3, its specific setting is absent in subsequent explanations. In Equation 9, where two loss functions are introduced, there is only one balancing factor, leading to reader confusion.
4. There are overall writing issues in the paper, including citation formatting and writing errors such as inconsistent tenses and mixed usage of abbreviations (e.g., Fig. vs. Figure, Table vs. Tab.). Careful proofreading and editing are required to enhance professionalism.
5. Figure 4 lacks a detailed explanation, making it challenging for readers to understand the purpose of the experiment and the impact of batch size on the results. More background information and clarification are needed.
I hope this feedback helps in further improving your research.

[1] Shadow Knowledge Distillation: Bridging Offline and Online Knowledge Transfer. 2022. In NeurIPS.
[2] Decoupled Knowledge Distillation. 2022. In CVPR.

**Strengths:**

See summary

**Weaknesses:**

See summary

**Questions:**

See summary

---

> ### Author Response · Authors · 2023-11-15
> **Responses to the review of Reviewer 9Nip-Part-I**
>
> We appreciate the valuable feedback provided for our work. Over the past few days, we have diligently worked to address all the concerns raised. ***We kindly request the reviewer to reconsider and potentially raise the score, and we have outlined our responses to each point below***. We will update the paper and appendices with the following your comments. Please feel free to let us know if you have any further concerns or questions to discuss.
>
> **Answer about Q1 and Q2)**
>
> Since both SHAKE and DKD are KL-divergence-based KD methodologies, our focus was on comparing cosine similarity distances with KL-divergence itself. KL-divergence is characterized by attempting to 'exactly match' the distribution of student predictions with the distribution of teacher predictions. This implies that the logit values of students and teachers must be 'exactly identical' to make the loss value zero. However, these matching criteria impose overly strict constraints on students, hindering their ability to acquire flexible dark knowledge from the teacher distribution and resulting in a decline in student performance.
>
> To address these constraints, we propose losses using cosine similarity. Cosine similarity possesses scale-invariant properties. Therefore, it is distinguished by providing flexibility to the logits of students without requiring them to be exactly the same as the teacher distribution. Consequently, the use of cosine similarity makes the acquisition of dark knowledge by the student distribution more flexible (as referred in Figure 2 of the main paper). We validated this through entropy analysis experiments (as referred in Figure 3 of the main paper), demonstrating that our method exhibits greater entropy compared to conventional KD methods. (It is important to note that higher entropy corresponds to a greater amount of dark knowledge.)
>
> That can be expressed using the following formula.
>
> KL divergence: $D=D_{KL}(p^{t}, p^{s})=0$ when $p^{t}=p^{s}$
>
> Cosine Similarity distance remains scale-invariance:
>
> $D=D_{cs}(p^{t}, p^{s})=0$ when $p^s=kp^t$, where $k>0$.
>
> ***Proofs of invariance of consine similarity***
>
> We demonstrate through the following equation that cosine similarity becomes scale-invariant.
>
> Consider that the student and teacher predictions are denoted as $p^s$ and $p^t$, respectively, and magnitude scaling on $p^s$ and $p^t$ can be formulated as $ap^t$ and $bp^s$, where $a×b>0$.
>
> $D_{cs}(ap^t, bp^s)=$$(ap^t)(bp^s)\over\vert ap^t \vert\vert bp^s \vert$=$\sum(ap^t)(bp^s)\over\sqrt{\sum(ap^t)^2}\sqrt{\sum(bp^s)^2}$=$ab\sum{(p^t)(p^s)}\over{a\sqrt{\sum{(p^t)^2}}b\sqrt{\sum{(p^s)^2}}}$=$D_{cs}(p^t,p^s)$
>
> ***Related Works***
>
> In addition, we will provide comparison with existing KD methods, DKD and SHAKE, as follows:
>
> DKD suggested dividing KL-divergence into TCKD and NCKD to minimize the suppression of dark knowledge (here, referred as NCKD) as the confidence of the teacher model in the training sample increases. In contrast, our approach addresses this challenge by introducing cosine similarity with scale-invariant properties. These scale-invariant properties alleviate constraints on the student model, enhancing its flexibility to acquire information from the teacher model. This, in turn, facilitates the acquisition of a more diverse range of dark knowledge.
>
> SHAKE generates an optimized model for student model by introducing a proxy teacher, departing from the traditional KD method of transferring knowledge from pretrained teachers to students. This approach aligns with our belief that conventional KL methods do not provide the most effective information to students. However, SHAKE requires additional training for the individual shadow head for this purpose, incurring an extra cost of approximately 1.28 times that of conventional KDs. On the other hand, our method can transfer optimal information without the need for extra training.
>
> Moreover, we aim to conduct further experiments by exploring mutual training between students and teachers using cosine similarity, similar to SHAKE. We intend to compare the results of these experiments with those of SHAKE to gain a comprehensive understanding of their effectiveness.
>
> We are currently working on acquiring further comparative analyses, including visualizations such as "tSNE" and "logits correlation", which we will incorporate.

---

> ### Author Response · Authors · 2023-11-15
> **Responses to the review of Reviewer 9Nip-Part-2**
>
> **Answer about Q3)**
>
> Initially, the balancing factor for 'CSWT' was set to 1, resulting in only one balancing factor in Equation 9. However, I will modify the formula as follows to incorporate your suggestion.
>
> $L_{Total}=L_{CE}+\alpha($$1\over{C}$$\sum_{j=1}^{C}L_{CSKD}(p^s, p^t, T))+\beta($$1\over{C}$$\sum_{j=1}^{C}L_{CSWT}(p^s, p^t, T_i))$
>
> The temperature $(T)$ applied to the CSKD loss was consistently set to 4 throughout the entire experiment, a commonly used value in other KD papers.
>
> Furthermore, CSWT is employed in conjunction with the CSKD loss in entire loss function, utilizing cosine similarity to ensure that the student model receives optimal dark knowledge. This involves applying different temperature scaling $(T_i)$ for each sample, dynamically adjusting the distribution according to the sample. Specifically, a small temperature is applied when the cosine similarity value between the student distribution and the teacher distribution is high, and a higher temperature is applied when it is low.
>
> **Answer about Q4)**
>
> Thank you for your careful and detailed comments; I will reflect them into main paper and the appendix.
>
> **Answer about Q5)**
>
> Our method is designed to leverage scale-invariant characteristics, providing the student model with richer dark knowledge from the teacher model. For comparing dark knowledge, specifically non-target class information, we utilized entropy, as illustrated in Figure 3 of main paper.
>
> In contrast to the previous KD method, which uses class distribution for one image as logits, we utilize the batch distribution for one class as logits. This choice is motivated by the fact that cosine similarity learns relational information rather than accurate value matching of the distribution. Using batch predictions allows us to obtain more diverse relational information, considering the constraint that class predictions have a total sum of 1.
>
> However, it's important to note that as the batch size increases, the number of relations that need to be satisfied also increases. Consequently, we infer that our method approaches KL-divergence-based KD with an exact matching property as the batch size increases. To illustrate this relationship with constraints, we present an entropy analysis in Figure 4 of main paper.
>
> In general, devices with limited memory that require knowledge distillation necessitate high performance at a small batch size. Therefore, our method holds a significant advantage over conventional KL methods.

---

### Meta-Review · Area_Chair_DXq8 · 2023-12-17

**Metareview:**

While the proposed KD method involving cosine similarity shows novel elements, there are weaknesses that have not been sufficiently addressed. Reviewers pointed out a lack of clear comparison with other KD methods, insufficient motivation and detail regarding the loss function, and unclear presentation of experimental results. The authors have made an effort to address some of these concerns, refining definitions and providing more information on their methodology. However, it appears that reviewers are not entirely convinced that the revisions sufficiently resolve the previously emphasized issues, particularly concerning the clarity of the methodology and potential bias introduced by batch-wise cosine similarity. Reviewers reached a consensus that the paper is not ready for acceptance to the conference, with an encouragement to the authors to refine the clarity of their methodology, address issues with batch dependency comprehensively, and conduct additional comparative analyses to reinforce their claims in future submissions.

**Justification For Why Not Higher Score:**

It appears that reviewers are not entirely convinced that the revisions sufficiently resolve the previously emphasized issues, particularly concerning the clarity of the methodology and potential bias introduced by batch-wise cosine similarity.

**Justification For Why Not Lower Score:**

N/A

---

### Decision · Program_Chairs · 2024-01-16

Reject